# Diagnosis of Synovial Chondromatosis of Temporomandibular Joint: Case Report and Literature Review

**DOI:** 10.3390/healthcare9050601

**Published:** 2021-05-18

**Authors:** Florent Destruhaut, Antoine Dubuc, Aurélien Bos, Laurent Fabié, Philippe Pomar, Jean-Claude Combadazou, Antonin Hennequin, Sara Laurencin

**Affiliations:** 1School of Dental Medicine, Paul Sabatier University, Toulouse’s Teaching Hospital (CHU Toulouse), 31062 Toulouse, France; dubuca@me.com (A.D.); abos.pro@hotmail.com (A.B.); philippepomar3@gmail.com (P.P.); jccombadazou@gmail.com (J.-C.C.); antonin.hennequin@free.fr (A.H.); laurencin.s@chu-toulouse.fr (S.L.); 2URU EvolSan, Paul Sabatier University, 31062 Toulouse, France; 3Oral and Maxillofacial Surgery, Private Practice, 31000 Toulouse, France; laurent.fabie@wanadoo.fr

**Keywords:** synovial chondromatosis, orofacial pains, temporomandibular disorders, temporomandibular joint

## Abstract

Synovial chondromatosis is a non-cancerous tumor characterized by the formation of multiple nodules of cartilage due to metaplastic development of the synovial membrane. Etiology can be a primary lesion, of which pathogenesis remains unknown, or low-grade trauma or internal disorders. This pathology can long remain undiagnosed and leads to therapeutic wandering, especially since clinical manifestations are non-specific. Symptoms may mimic temporomandibular disorders and dental orthopantomogram does not always show intra-articular calcified bodies. Cone beam computed tomography (CBCT) and magnetic resonance imaging (MRI) are tests of choice for the diagnosis of this pathology. This case report describes the clinical manifestations, diagnosis and management of a case of synovial chondromatosis involving the temporomandibular joint, in a 21-year-old woman who was initially treated for two years for a common temporo-mandibular disorder. The evidence gathered during the medical interview and clinical examination led us to suspect synovial chondromatosis of the temporomandibular joint. Prescription of a CBCT and MRI confirmed the diagnosis of her temporomandibular joint disorder and led to a successful arthroplasty.

## 1. Introduction

Synovial chondromatosis (SC) is described as a non-cancerous tumor arising in the lining of a joint with formation of benign nodular cartilaginous proliferations from the synovial tissue. These nodules may loosen inside the articular space and may cause chronic pain and swelling in the preauricular region when the temporomandibular joint is involved [1]. SC mainly affects large joints (knee, hip, shoulder or elbow), and localization in the temporomandibular joint (TMJ) is a rare finding that was first reported in 1558 by Ambroise Paré [2]. Both upper and lower compartments of the TMJ can be affected, but Sozzi et al. underline the fact that very few publications are available about lower compartment SC, showing that these localizations are extremely rare [3]. When not confined in the articular space, extra-articular extensions may occur in the infratemporal space, parotid region, or middle cranial fossa [4,5].

Synovial chondromatosis was first classified in three classes by Milgram in 1977.

Early stage (Class I) is characterized by synovial symptoms without any loose bodies visible on X-Rays.Intermediate stage (Class II) reunites both clinical symptoms and images of calcified loose bodies within the synovial membrane.Final stage (Class III) is characterized by calcified loose bodies without any synovial involvement [6].

This classification was completed by Gerard et al. in 1993, with criteria based on the calcification degree of the loose bodies [7]. Then, in 2012, Chen et al. provided a classification for SC of the lower compartment of the TMJ, based on the different structures of the joint involved [8].

Differential diagnosis between temporomandibular disorders and synovial chondromatosis may be difficult and challenging, especially when the patient is young. In this context, we report the case of a synovial chondromatosis of the TMJ, in a 21-year-old woman, which caused intense orofacial pain. Practical recommendations, in terms of clinical and radiological examination, to avoid diagnostic and therapeutic wanderings, are also given.

## 2. Case Presentation

A 21-year-old woman came to the general dental consult of Toulouse’s Teaching Hospital (France) for localized pain involving her left temporomandibular joint. She reported intense, persistent pain, increasing during chewing and/or muscle fatigue, associated with joint noises described as “squeaking”. She explained being followed for the last 2 years for osteoarthritis of the left temporomandibular joint associated with myalgia. For this, the patient had benefited from maxillofacial physiotherapy sessions and the use of an occlusal splint associated with a behavioral approach in the context of her chronic pain. Despite the treatment, she showed no improvement and consulted for further advice in our dental hospital. The semi-directive interview revealed a history of whiplash that had occurred several years earlier, without head trauma. Clinical examination, based on Diagnostic Criteria for Temporomandibular Disorders (DC-TMD) [9], revealed pain during the external palpation of the left temporomandibular joint, in a closed mouth position, and during the retro-condylar palpation in a wide-open mouth position. Auscultation of the TMJ with a stethoscope revealed “squeaking type” noises. Examination of the mandibular kinematics showed a limitation in the opening of the mouth (31 mm) and a left mandibular deviation. The clinical features tended to confirm the diagnosis of osteoarthritis of the left TMJ, which was unlikely, however, given the patient’s young age, thus leading us to prescribe additional radiological examinations.

The dental orthopantomogram was not very contributive regarding lesions of the left condyle. No joint pinching, enlargement, or lysis of the condylar head was found ruling out the unlikely diagnosis of osteoarthritis (Figure 1). However, a discrete radiolucent image was visible in front of the left condyle, which could have been interpreted as a radiological artifact. A CBCT was prescribed and was able to highlight several radio-lucent bodies indicating the presence of calcified bodies typical of synovial chondromatosis (Figure 2). The diagnosis of synovial chondromatosis led to an arthroplasty of the pathological temporomandibular joint. This procedure allowed the removal of the calcifications in the synovial lining as well as inflamed joint tissue and synovial metaplasia. This surgical procedure was associated with a mandibular condyloplasty and a temporal condylectomy.

About thirty chrondroid fragments varying in size from 1 to 7 mm (the largest one measuring 7 × 5 mm^2^) were removed from the synovial lining. Anatomopathological analysis carried on these calcified bodies confirmed the diagnosis of synovial chondromatosis. These nodules are composed of a locally mixoid cartilaginous framework and punctuated with calcium deposits. Some bodies were developed under the synovial intimal layer, thus confirming the diagnosis of class II of Milgram.

Following the surgery, maxillofacial rehabilitation sessions (physiotherapy and occlusal splint) were prescribed in order to recover correct oral opening and ensure the normalization of occlusal contacts.

## 3. Discussion

Etiologies of synovial chondromatosis are still unknown and are discussed in the few publications on the subject. Suspected etiologies can be divided into two categories:Primary lesions, of which pathogenesis remains unknown. Cartilaginous metaplastic proliferation of fibroblasts is a hypothesis and could cause primary lesions, without any well-identified genetic or risks factors [10].Secondary Synovial chondromatosis following, infection, embryological disturbances or trauma such as repetitive low-grade trauma, internal derangement, parafunctions, or even degeneration [11,12]. Synovial chondromatosis could develop in a few years following whiplash [13], and signs of joint pain and noises associated in young adults should alert the practitioner, as in the clinical case presented here.

The study of the recent scientific literature gives no precise prevalence of synovial chondromatosis of the temporomandibular joint. In general, SC affects men more frequently than women except for the temporomandibular joint which is more frequently affected in women with a 4:1 sex ratio [13]. Diagnosis of synovial chondromatosis of the temporomandibular joint must therefore be systematically sought in the presence of persisting pain, in a young female patient with no signs of improvement following common temporal-mandibular disorder treatment. Clinical diagnosis of SC is all the more difficult because symptoms often combine preauricular swelling and chronic unilateral pain, associated with limitation or modified jaw motion, crepitus, and occlusal changes [12]. These symptoms mimic TMJ disorders, the prevalence of which is more important (17% at 19–20 years and 10% at 30–31 years) [14]. Since these clinical signs are non-specific, and similar to those of internal disorders or arthritis, delayed diagnosis is frequent. In the case reports described by Benslama et al., the average delay in diagnosis was 11 months, and it could even reach 31 months according to Ardekian et al. [15,16]. Considering these more frequent diseases do not require any particular TMJ imaging, their diagnosis being based on the clinical examination, practitioners do not systematically prescribe complementary X-Rays or MRI examinations [17]. This could explain why patients suffering from synovial chondromatosis of the TMJ wander from lack of early positive diagnosis as far as therapeutic management is considered.

Nevertheless, as soon as Cone Beam-Computed Tomography (CBCT), CT-scan or Magnetic Resonance Imaging (MRI) are performed, loose calcified and/or non-calcified bodies can easily be identified, with or without extra-articular extensions [18,19]. Lee et al. recommend the use of MRI as a routine procedure to avoid misdiagnosis. MRI can detect all non-calcified cartilage free bodies and is usually the complimentary exam of choice in the early stages of the disease especially since CT-scans can only detect calcified free bodies [20].

Diagnostic wise, TMJ arthroscopy is a real improvement and can also be used for treatment [21]. Brabyn et al. led a systematic review of case reports about patients treated with an arthroscopic approach between 2007 and 2017, concluding that development of CT, MRI imaging, and examination through arthroscopy allowed earlier diagnosis of SC [22]. Moreover, Fernández Sanromán et al. concluded, relying on a case series of five patients, that arthroscopy should become the option of choice as far as treatment is concerned in SC limited to intra-articular localizations [5]. Cai et al. led the largest case series about SC of the TMJ in the literature, and came to the same conclusion, showing good results with this technique [21]. The only limit is the diameter of the foreign bodies (if they are larger than the diameter of the arthroscopy cannula). However, free bodies up to 6.5 mm in diameter can be removed arthroscopically.

Open surgery is indicated in case of invasive SC into adjacent tissues (middle cranial fossa, ear and carotid canal) or if free bodies are present in the lower compartment [23,24,25,26] (Figure 3).

Although diagnosis on the basis of clinical elements can be difficult and confusing with TMJ disorders, the therapeutic approach is, at present, widely proven. Open arthrotomy with removal of loose bodies is the gold standard in terms of treatment for this disease. Synovectomy, with or without condylotomy or condylectomy can be associated to this procedure if necessary [6]. Oral–facial rehabilitation, through physiotherapy and/or occlusal splint, should be conducted immediately after surgery to allow normalization of functions (chewing, oral opening). Even though CBCT or MRI facilitate the diagnosis of SC, the surgery by enabling histopathological analysis of the removed bodies is essential to confirm the diagnosis [14].

Finally, the recurrence rate seems to be low. Cases of recurrence have been described and have required larger resections that can go as far as condylectomy and require reconstruction [17,27].

Concerning the clinical case presented, the one-year clinical follow-up of the patient showed clear improvement of the oral opening and the absence of painful symptoms. In addition, the presence of a Milgram stage 2 requires radiological monitoring for possible recurrence due to the possible persistence of metaplastic foci.

## 4. Conclusions

Synovial chondromatosis is a relatively rare pathology when it is located at the temporomandibular joints. Clinical signs being non-specific, it can be mistaken for more common temporomandibular disorders whose prevalence are greater. The appearance of pain in the temporomandibular joint with an oral opening limitation in young adults, and more particularly in young women, must alert the clinician, especially if whiplash, even several years back, is mentioned during the anamnesis. Cone beam-computed tomography or magnetic resonance imaging will make it possible to make an early and accurate diagnosis, thus avoiding therapeutic wandering and implementing appropriate management with maxillofacial surgery centers.

## Figures and Tables

**Figure 1 healthcare-09-00601-f001:**
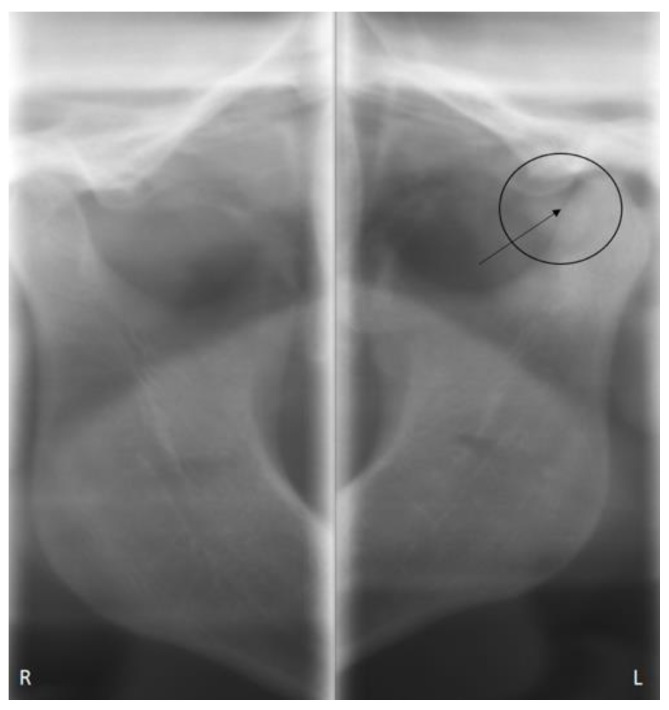
OPT = Orthopantomography.

**Figure 2 healthcare-09-00601-f002:**
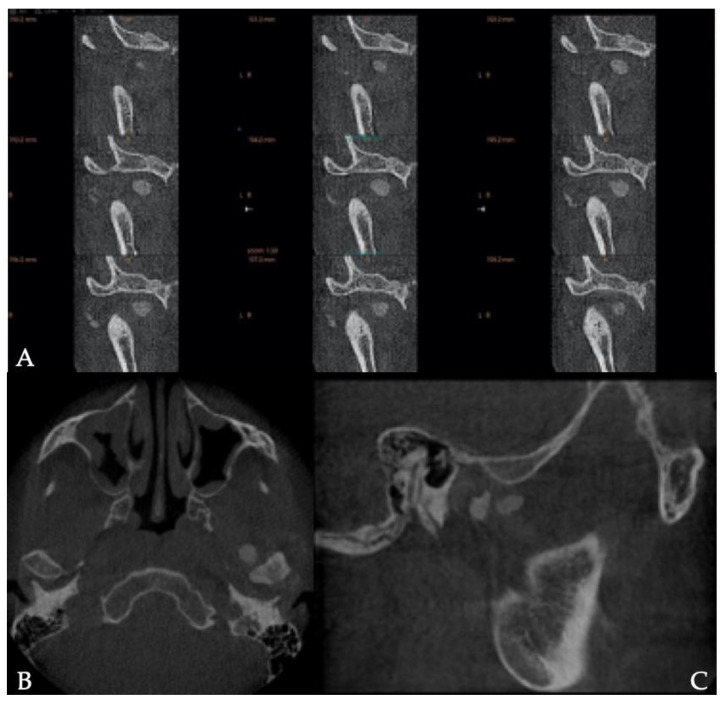
Coronal(**A**), axial (**B**) and sagittal (**C**) sections of the temporo-mandibular joint extracted from the CBCT: presence of calcified bodies around the condyle.

**Figure 3 healthcare-09-00601-f003:**
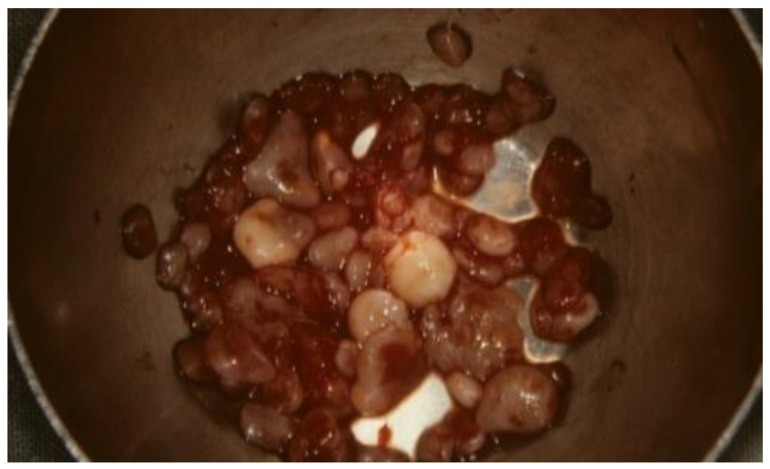
Intraoperative view of calcifications. Some calcified bodies are surrounded by synovial metaplasia.

## Data Availability

Not applicable.

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
