# Peer review of "Diagnosis of Synovial Chondromatosis of Temporomandibular Joint: Case Report and Literature Review"

_healthcare, 2021, doi:10.3390/healthcare9050601_

Round 1
Reviewer 1 Report
I appreciate the opportunity to review a case report entitled “Diagnosis of Synovial Chondromatosis of Temporomandibular Joint: Report of a Case and Literature Review ”. Synovial Chondromatosis is a rare disease. We are falling behind from having the whole picture of the disease. The accumulation of each case, therefore, will be important for giving the best treatment to patients in the future. I put some points I have noticed below at this time. I think that some data and descriptions should be added. Hope this helps.
- The three direction (axial, coronal and sagittal plane ) images of CT and MRI are required. There is no need for several slices of one direction(fig.2).
- The operative arthroscopic view showing loose bodies and synovial membrane(synovial metaplasia?) should be added.
- The pathological appearance of loose body and synovial membrane (synovial metaplasia?) should be added.
- How is the grade of your case in Milgram classification and Gerard classification?
- Inferior joint space was unaffected in your case, wasn’t it?
- Please discuss recurrence in terms of the rate, surgical method, its duration after initial treatment and the like.
Author Response
We would like to thank the reviewers for the time they spent on their careful examination of our manuscript and their very constructive comments and suggestions.
Our responses to the reviewers are noted point by point below. We give our explanation in response to each comment below, together with the changes made. These changes are printed in red, both here and in the manuscript.
We hope that the modifications made will be satisfactory and will enhance the quality of the manuscript.
Please see the attachment.

Reviewer 2 Report
The manuscript from Destruhaut et colleagues describes a case report of a patient suffering from synovial chondromatosis of the TMJ. The work is well written and well organized. Clinical approach has showed an interesting effort, and it seems to reach a notable scientific accuracy. Nevertheless, this reviewer major concern is about first diagnostic pathway: why didn't authors conceive a TMJ magnetic resonance as first step exam? Usually, MR is considered as the gold standard assessment for TMJ disorders, thus it is also a less invasive procedure.
Author Response

(The authors gave the same response as above.)

Round 2
Reviewer 1 Report
Thank you for submitting the revised manuscript. Some changes for the better were given in the revised version. However, there is still a couple of room for improvement. More information is better in case reports for readers who will diagnose and treat the diseases.
- Didn’t you take the MRI images? Three direction images of MRI should be indicated.
- I suggested that the operative arthroscopic view showing loose bodies and synovial membrane be given. Figure 3 showed the resected loose bodies, not operative arthroscopic views. Did you perform arthroscopic surgical treatment, as described in “ Abstract”? I get confused as to whether you did arthroscopic surgical treatment or open arthrotomy because of the legend of Figure 3 in the original version, which said, “Calcified bodies after open surgery”. The treatment method is very important, You, therefore, should describe it precisely and minutely.
- Hadn’t you examined the resected specimens pathologically? You described the importance of the histopathological analysis of removed bodies on page 6. The pathological appearance should be indicated.
Author Response
We would like to thank the reviewers for the time they spent on their careful examination of our manuscript and their very constructive comments and suggestions.
Our responses to the reviewers are noted point by point below (attached files). We give our explanations in response to each comment below, together with the changes made. These changes are printed in red and highlighted in yellow, both here and in the manuscript.
We hope that the modifications made will be satisfactory and will enhance the quality of the manuscript.
